| Open Peer Review | Clinical Microbiology | Methods and Protocols

# Targeted amplification-based whole genome sequencing of *Monkeypox virus* in clinical specimens

S. Isabel,[1] A. Eshaghi,[1] V. R. Duvvuri,[1,2] J. B. Gubbay,[1,2] K. Cronin,[1] Aimin Li,[1] M. Hasso,[1] S. T. Clark,[1] J. P. Hopkins,[1,3] S. N. Patel,[1,2] T. W. A. Braukmann[1]

**ABSTRACT**   The 2022 mpox outbreak has led to more than 91,000 cases in 115 countries. Whole genome sequencing (WGS) has been at the forefront of surveillance and outbreak investigations for different pathogens of public health significance. Many institutions performing WGS on *Monkeypox virus* (MPXV) use a resource-intensive metagenomic approach. Here we present a targeted amplification method for WGS of MPXV from clinical specimens. We designed 43 pairs of primers (amplicons ~5 kb) with PrimalScheme to span the ~200 kb viral genome and then added 12 additional primers to optimize amplification. We extracted nucleic acid from clinical specimens and amplified the two primer pools. All libraries were sequenced on the MiniSeq platform. Resulting reads were filtered by quality and then mapped to a MPXV reference genome. Consensus sequences were generated for phylogenetic analysis. A total of 91 specimens with a real-time-PCR cycle threshold (Ct) values ≤27.9 were sequenced using our targeted amplification protocol. The sequenced MPXV genomes were of high quality with mean genome coverage of 99.56% (95% CI 99.32-99.80%), mean depth 1,395× (95% CI 1275–1515), and mean mapping quality of 52.87 (95% CI 52.1–53.6) and allowed for greater multiplexing of samples relative to metagenomics. The MPXV genomes belong to 8 of the 13 clades observed during the 2022 global mpox outbreak. Targeted amplification enrichment provides high coverage, throughput, and short turnaround times. It is an efficient low-cost method for MPXV WGS and can benefit public health surveillance and outbreak management.

**IMPORTANCE**   We present a protocol to efficiently sequence genomes of the MPXV-causing mpox. This enables researchers and public health agencies to acquire high-quality genomic data using a rapid and cost-effective approach. Genomic data can be used to conduct surveillance and investigate mpox outbreaks. We present 91 mpox genomes that show the diversity of the 2022 mpox outbreak in Ontario, Canada.

**KEYWORDS**    *Monkeypox virus*, mpox, WGS, targeted amplification

In 2022, the World Health Organization (WHO) reported an outbreak with currently more than 91,000 confirmed mpox cases in 115 countries among all six WHO regions (1). Mpox is a disease caused by the *Monkeypox virus* (MPXV), a double-stranded DNA virus in the *Orthopoxvirus* genus, which also includes the *Variola virus*, the causative agent of smallpox (2). Mpox often presents with a skin rash and/or mucosal lesions and other systemic symptoms (e.g., fever, lethargy, lymphadenopathy, myalgia, and headache) and can cause death (3, 4). The modes of transmission of mpox include direct and indirect contact, respiratory droplets, and animal-to-human contact (3). The 2022 mpox outbreak is unique because it represents the first global sustained human-to-human spread of MPXV. Phylogenetic analysis of MPXV genomes implicated subclade IIb

Address correspondence to T. W. A. Braukmann, Tom.Braukmann@oahpp.ca.

S. Isabel and A. Eshaghi contributed equally to this article. Author order was determined by a rock, paper, scissor contest.

J.B.G. was a paid consultant scientific editor for GIDEON Informatics, Inc., which was unrelated to the current work.

in the recent global spread of mpox (5, 6). In Canada, 1,512 mpox cases were confirmed, including 722 in the province of Ontario, as of 29 September 2023 (7, 8).

Genomic epidemiology, the use of genome sequences to study infectious disease transmission and evolution, has transformed how we manage and respond to disease outbreaks (9, 10). Genomic sequences enable surveillance programs to track circulating and emerging variants (11–15), detect drug resistance (16), improve contact tracing (17), map transmission dynamics (17, 18), and design novel vaccines (9). Salvato et al. (19) found evidence of transmission between a patient and a healthcare worker based on identical MPXV whole genome sequences, which led to novel mpox-specific measures to control viral spread in healthcare settings. Genomic surveillance programs have the potential to enhance public health responses with improved genomic data quality and analytics. However, the success of genomic epidemiological programs is dependent on providing rapid and cost-effective sequencing solutions that can deliver results within an actionable time frame.

Most institutions performing whole genome sequencing (WGS) on MPXV use a resource-intensive metagenomic approach that sequences all nucleic acids within a sample and requires more sequences to recover a target genome (16–25). Targeted amplification-based WGS approaches have proven cost efficient and effective at recovering WGS for viruses with smaller genomes such as SARS-CoV-2 (~30 kb), Zika virus (~11 kb), and Ebola virus (~19 kb). Targeted amplification protocols for these viruses have used amplicon sizes ranging from 400 to 2,500 bp (20–24). The MPXV viral genome differs from those mentioned above in relation to its nucleic acid composition (DNA), size (~200 kb), and complexity (contains long stretches of repetitive homopolymer sequences), which complicates the design of a targeted amplification-based WGS assay.

We designed a targeted amplification-based WGS protocol to sequence MPXV genomes for surveillance and outbreak investigation of mpox. Our approach uses longer amplicons (~5 kb) to compensate for longer genome and homopolymer regions of MPXV relative to viruses with smaller genomes.

## MATERIALS AND METHODS

### Sample selection and PCR

This retrospective study conducted MPXV WGS directly on patient specimens (lesion, throat/oral, genital region or lesion, or nasopharyngeal swabs) in universal transport media (Copan, USA). Specimens were heat inactivated at 60°C for 1 hour (25) and nucleic acids were extracted using the NucliSENS EMAG system following manufacturer's instructions (bioMérieux Canada Inc, St-Laurent, Quebec, Canada). Multiplex MPXV real-time (rt)-PCR using a clade II-specific G2R/G2L region (*TNFR* gene) and an RNaseP control was performed as previously described (26, 27). Positive specimens with the lowest rt-PCR cycle threshold (Ct) values for each patient collected from June to September 2022 were sequenced retrospectively as part of a convenience sampling strategy. Specimens with target rt-PCR Ct values of 30 or more were rejected due to presumed lower viral loads.

### WGS primer design

We used PrimalScheme (24) under default parameters to generate an MPXV primer scheme using a reference sequence (GenBank ON563414.3) from the recent global outbreak. The primers were designed to bind outside of homopolymer, repetitive, and/or hypervariable regions. The primers were multiplexed into two PCR pools, creating a tiled amplification of ~5 kb with an overlap of a minimum of 300 bp and spanning the entire MPXV genome. Initially, 43 primer pairs (86 primers) were designed. Subsequent PCR testing indicated the need for primer concentration adjustments and for 12 additional primers to fill the low coverage or missing regions for a total of 98 primers. The final primer scheme for pools 1 and 2 included 46 and 52 primers, respectively (Table S1).

## Genome PCR amplification

Specimens with low rt-PCR Ct values (<18) based on the diagnostic PCR were diluted in nuclease-free water to increase their theoretical rt-PCR Ct values above 18 to prevent inhibition in downstream procedures. The viral DNA was subjected to two PCRs, containing pool 1 and pool 2 primers (Table S1). Each PCR of 25 µL included the following: 5 µL of Q5 Hot Start buffer (New England Biolabs, Ipswich, MA, USA), 0.5 µL of 10 mM dNTP, 0.5 µL of Q5 High-Fidelity DNA Polymerase, 1 µL of primer pool 1 or 2, 5 µL of template DNA, and 13 µL of PCR grade water. The following thermal cycling conditions were used on an ABI SimpliAmp thermocycler (Applied Biosystems, Waltham, MA, USA): initial denaturation at 98℃ for 2 min, followed by 45 cycles at 98℃ for 10 seconds and 65℃ for 5 min (with an increment of 10 seconds per cycle), concluding with the final extension at 72℃ for 5 min. The presence of pooled amplified products was confirmed through electrophoresis on 1% agarose gel.

## Library preparation

Equimolar amounts of each amplicon pool were mixed and cleaned with AMPure beads (0.5 ratio). For library preparation, we used 0.5 ng of purified amplicons using Nextera XT DNA Library preparation kit (Illumina Inc., San Diego, CA, USA) in half volume, according to the manufacturer's recommendation. The normalized library (1.2 pM) was sequenced using an Illumina MiniSeq system (Illumina Inc.) with High Output Reagent Kit (300 cycles) following the manufacturer's recommendation.

## MPXV genome assembly

Read quality was assessed with fastqc (v0.11.9) (28). Reads were then filtered and trimmed for quality using fastp (v0.20.0) (29) under default settings. Filtered reads were then aligned to the MPXV genome reference MT903344.1 using minimap2 (v2.24.r1122) (30, 31). Variants were called using iVar (v1.3.1) (32) with a minimum depth of 50 and quality of 30, and a minimum frequency of 0.7. iVar (v1.3.1) (32) was also used to call consensus sequence with a minimum quality of 30, minimum depth of 10X, and a frequency threshold of 0.7. Genome coverage and depth statistics were collected using the Samtools depth function (33).

To place Ontario MPXV in the global context, consensus genomes were combined with sequences representing both clades I and II, along with a sampling of Canadian sequences from GenBank and used by Nextstrain (34) (Table S2). Sequences were aligned using mafft (v7.50.8) (35) with the 6merpair algorithm and the exclusion of insertions (--keeplength parameter). Coverage plots were generated in R (v.4.2.2) (36) using the tidyverse (37), ggplot2 (38), and reshape2 (39) packages. Descriptive statistics and Spearman's rank correlation analysis were performed using GraphPad Prism (v9.5.1) (40).

## Phylogenetic analysis

A maximum likelihood tree was generated using IQTree2 (41) (v1.6.2) with a model finder to select the best substitution model. Branch support was estimated by 1,000 ultrafast bootstrap replicates (41, 42).

## RESULTS

## Targeted amplification enrichment

We introduced a targeted enrichment method using a multiplex tiling PCR to amplify the ~200 kb MPXV genome using long amplicons (~5 kb). Optimization and adjustment of the protocol were done by introducing additional primers to obtain adequate amplification and coverage in certain genome regions (including the inverted terminal repeats) that had lower read depth (Table S1). Furthermore, a long extension time of 5 min at 65℃, accompanied by an increment of 10 s/cycle, was needed to

optimize amplification. Our targeted amplification protocol recovered near-complete MPXV genomes with high depth and coverage (Fig. 1).

## Whole genome sequencing

A total of 95 specimens were submitted to PCR amplification for sequencing. Four did not amplify during targeted enrichment with prior diagnostic rt-PCR Ct values between 21.6 and 27.8. These four specimens were re-tested with the diagnostic rt-PCR (26, 27): one had rt-PCR Ct value of 28.3 and re-amplified for sequencing successfully; three had Ct values between 30.6 and 31.2, suggesting possible DNA degradation during conservation.

The 91 clinical MPXV specimens from unique patients sequenced represent 14% of the 644 PCR-positive mpox patients in the four-month study period. We included the following specimen types and counts: 63 lesions, 24 genital area or genital lesions, 2 oral, 1 nasopharyngeal, and 1 not specified swabs (rt-PCR Ct values ranging from 10.8 to 27.9). The targeted amplification method provided high genome coverage (median 99.80%; mean 99.56%, 95% CI 99.32–99.80%), high depth (median 1336; mean 1395; 95% CI 1275–1515), and median mean mapping quality of 53.7 (95% CI 52.1–53.6) (Fig. 1 and 2; Table S3). All samples had a mean depth greater than 100× but one sample (PHOL149) had a mean depth close to 100× (110×; Table S3). All samples were equimolar pooled prior to sequencing. Variation in sequencing depth can be a result of an overestimated library concentration, pipetting error, or a combination of both. However, all samples still had a relatively high read depth (100× or greater).

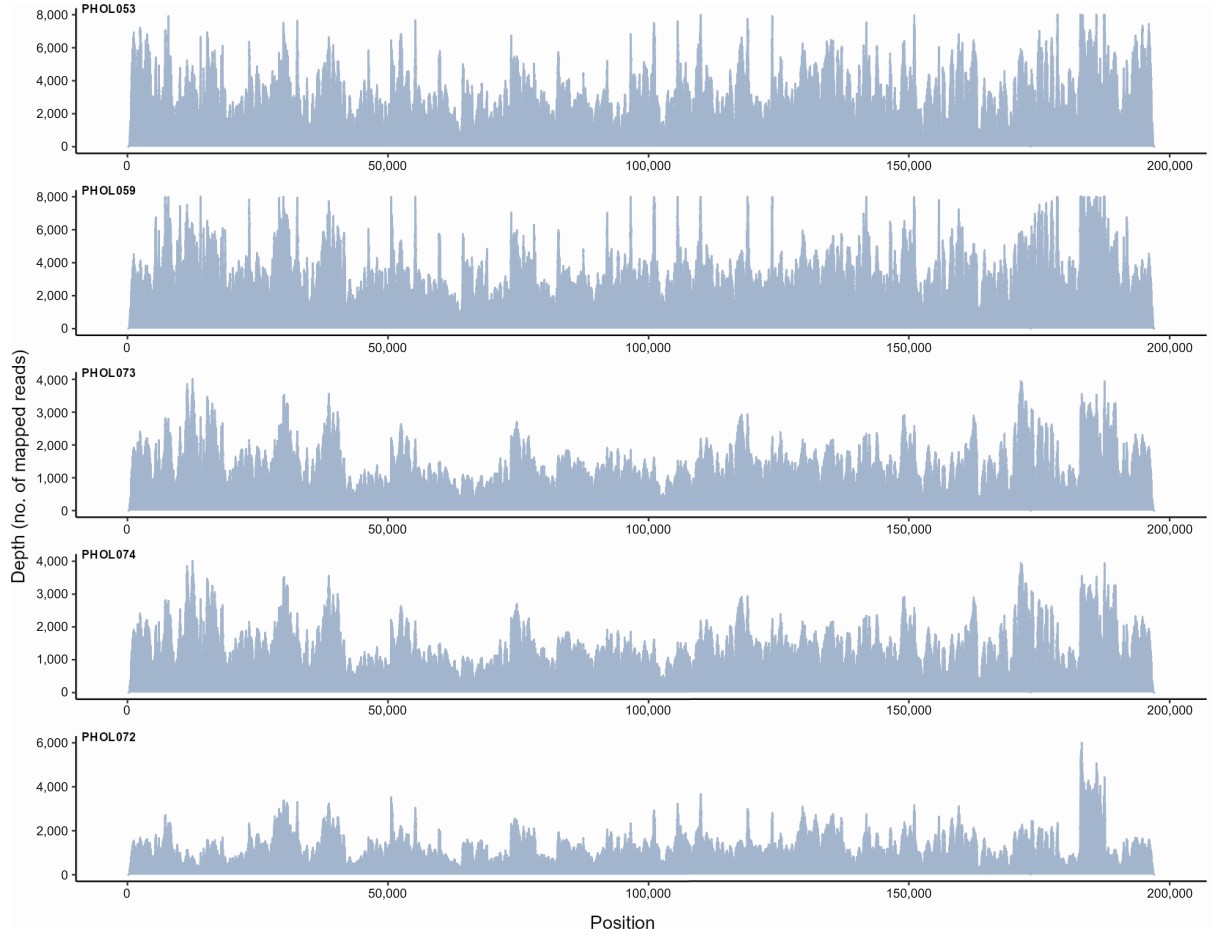

**FIG 1** Schematic representation of a coverage plot across the *Monkeypox virus* genome. Five representative specimens (PHOL) with diagnostic rt-PCR cycle threshold values between 16 and 22 are presented.

We sought to investigate the potential correlation between the rt-PCR Ct values (surrogate for viral loads) and WGS efficiency. MPXV rt-PCR Ct values ranged from 10.8 to 27.9. Seven of the 91 specimens had associated rt-PCR Ct values between 25.0 and 27.9 and yielded high genome coverage (median 99.66; mean 98.40%; 95% CI 95.31–101.5%) and mean depth (median 1129; mean 1043; 95% CI 482.5–1604). Additionally, we did not find a correlation between the rt-PCR Ct values and genome coverage (Spearman's rank correlation coefficient, r −0.03012, 95% CI −0.2404 to 0.1829, $P$ value 0.7768) or mean depth (Spearman's rank correlation coefficient, r −0.1376, 95% CI −0.3396 to 0.07647, $P$ value 0.1933) (Fig. 2). The turnaround time, including sample preparation, library preparation, sequencing, and bioinformatics, is less than 48 hours. This method allows for multiplexing of up to 30 specimens on a single MiniSeq flow cell.

## Phylogenetic analysis

A maximum likelihood tree was constructed using IQTree2 under an HKY + I + F model selected using the Bayesian Information Criterion (43). The phylogenetic tree contained 91 MPXV genome sequences from Ontario generated along with strains representing the global diversity of MPXV genomes (Fig. 3; Fig. S1; Table S2). The phylogenetic analysis showed MPXV genomes from Ontario sequenced in this study belong to clades B.1, B.1.1, B.1.2, B.1.3, B.1.4, B.1.7, B.1.8, and B.1.12.

## DISCUSSION

Here we present a long amplicon (~5 kb) multiplex PCR genome amplification protocol for WGS of the ~200 kb MPXV. This protocol achieves genomes with high coverage and mean depth across MPXV clade II sequences. Although this assay was designed at the beginning of the outbreak when a small number of sequences were publicly available, amplification was still effective over the study period with genomes recovered from eight MPXV clades. Our targeted amplification strategy used tiled amplicons, which is superior to metagenomics for sensitivity and specificity because it selectively amplifies target pathogen sequences.

The reduction or elimination non-target reads achieved by targeted amplification simplifies bioinformatics analysis (23, 24) and enables increased sample multiplexing relative to a metagenomic or bait capture methods as a larger percentage of reads are for the target pathogen. Increased sample multiplexing reduces sequencing costs per sample, and library preparation costs are cheaper than bait-capture methods, an alternative method for enriching target pathogen reads (44). Hybrid bait-capture often produces more even coverage across genomes but has greater host contamination, is a more laborious method, and has a greater variation between batches than targeted amplification (44). Due to its low cost and high multiplexing potential, targeted amplification using tiled amplicons is increasingly common for viral WGS directly from clinical

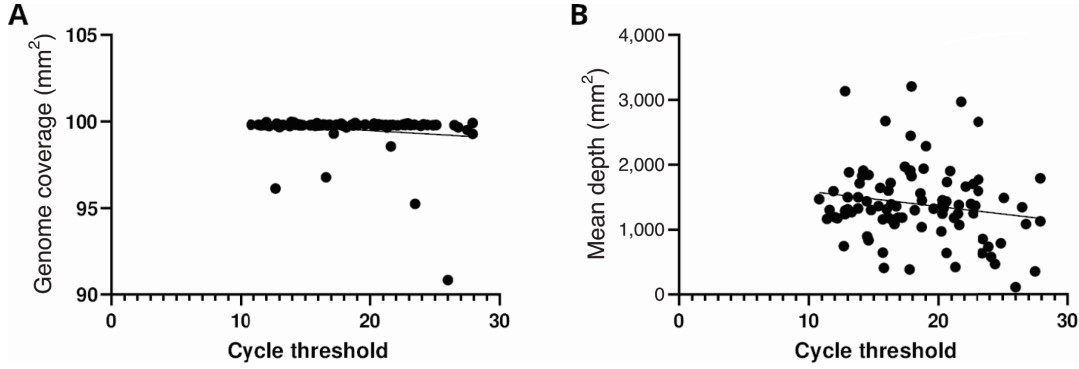

**FIG 2** *Monkeypox virus* whole genome sequencing data compared to rt-PCR cycle thresholds. (A) Genome coverage (%) and (B) mean depth (number of mapped reads) are presented with lines of best fit.

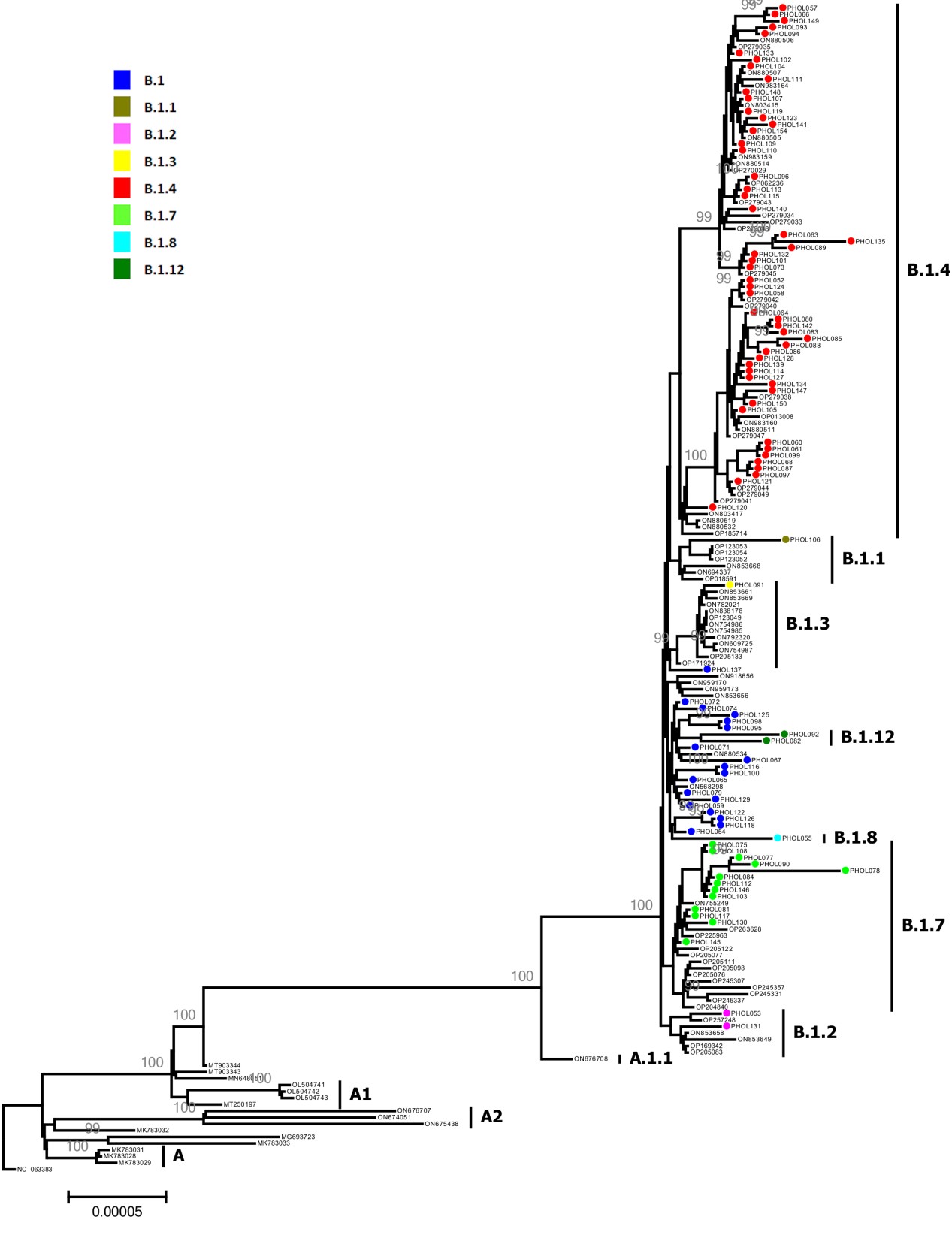

**FIG 3** Maximum-likelihood tree of WGS of *Monkeypox virus* specimens from Ontario and references. Phylogenetic trees were rooted using NCBI reference sequence NC_063383. A total of 195,933 positions were included in the final data set. The scale bar represents the number of nucleotide substitutions per site. Lineages are indicated with colored circles defined above. Colors denote specimens sequenced in this study belong to different MPXV clades. A version of this tree with bootstrap values is available in Fig. S1.

specimens. The choice of any tiled amplicon method needs to be adjusted and optimized to meet the analytical sensitivity and genome complexity required for the pathogen and specimen types in question (21). However, one caveat with primer-based approaches is that mutations occurring in priming sites can lead to amplicon drop out, requiring new primers to be designed. This can be problematic for viruses with high mutation rates, such as SARS-CoV-2 and influenza. However, both these viruses are routinely sequenced using amplification-based approaches (20, 45, 46).

Recent studies have shown that amplicon-based WGS of MPXV provides higher genome coverage compared to metagenomics (19, 47–50). These protocols have used multiplexed PCR on clinical specimens with amplicon lengths ranging from 1,500 to 2,500 bp, smaller than our ~5 kb approach (19, 47, 50). For example, Chen et al. (50) designed an amplicon-based enrichment with a scheme of 163 amplicons (1597–2497 bp), which is more complex than our scheme of 49 amplicons (50). Amplicon-based approaches recover near-complete genomes (19, 47, 50), similar to the near-complete genomes recovered in this study. We used a long amplicon approach for our tiling-based protocol because fewer primers decreases the likelihood of primer interactions that can interfere with PCR amplification (24) and are less prone to mutations leading to amplicon drop out. Longer amplicons are more suitable for recovering genomes from viral pathogens with longer genomes such as MPXV; longer amplicons are also more suitable for long read sequencing platforms that can bypass fragmentation prior to sequencing. An advantage of our long amplicon protocol is that we circumvent binding issues at the 5′ and 3′ end of the genome that contain inverted terminal repeats that are observed with protocol using shorter amplicons (19, 47, 50).

To be practical, any longer amplicon protocol needs to maintain sensitivity across a range of microbial loads and clinical specimen types. In a prior study conducted at our institution, MPXV rt-PCR Ct values from all types of specimens ($n = 1033$) ranged from 11.1 to 39.6; specimens with the lowest mean Ct values were skin lesions and throat swabs with 23.1 and 27.8, respectively (26). These Ct values suggest high viral loads in many collected MPXV-positive specimens. In our study, we performed MPXV WGS on a variety of specimen types. Based on our selection criteria and amplicon pools, our long amplicon approach was sufficiently sensitive to recover near-complete MPXV genomes from 95.8% of specimens (91/95) with rt-PCR Ct values (<30). Our protocol did not establish the limit of detection of the MPXV WGS assay, but this method is ideal for generating mpox genomes used for surveillance, outbreak investigation, or research. Further validation is needed to determine the sensitivity of the assay, but it can reliably generate genomes of specimens with rt-PCR Ct values <30 (Table S3).

Here, we showed we produced high genome completeness for eight different clades of MPXV. The phylogenetic tree shows that the 91 specimens from Ontario are from eight of the thirteen clades observed in the global 2022 MPXV outbreak. These specimens accounted for only 14% of the PCR-positive mpox cases in the study period. This suggests that multiple importations led to the mpox outbreak in the province, and this pattern of transmission is consistent with multiple importations seen in other countries during the global outbreak (2). Further study with associated epidemiological data would be required to understand the importations from other jurisdictions and transmission in Ontario.

## Conclusion

Our targeted amplification using a tiled strategy with PCR enrichment provides high genome coverage and throughput, reduces turnaround time compared to metagenomic approaches, and can benefit public health surveillance and outbreak management. To our knowledge, this is the first study presenting an amplification-based enrichment WGS using a 5-kb amplicon with a variety of clinical specimens. We showed that targeted amplification enrichment is an efficient method for the large viral genome (~200 kb) of MPXV. Our study paves the way for targeted enrichment WGS of other viruses with large

genomes such as Epstein–Barr virus (~170 kb) and human cytomegalovirus (~235 kb) that have important health implications and encompass antiviral resistance (51, 52).

## ACKNOWLEDGMENTS

We thank the scientists who have publicly shared MPXV genome data. We are thankful to the molecular diagnostics team at Public Health Ontario for providing technical support.

## AUTHOR AFFILIATIONS

[1]Public Health Ontario Laboratory, Public Health Ontario, Toronto, Ontario, Canada
[2]Department of Laboratory Medicine and Pathobiology, Temerty Faculty of Medicine, University of Toronto, Toronto, Ontario, Canada
[3]Department of Health Research Methods, Evidence, and Impact, McMaster University, Hamilton, Ontario, Canada

## AUTHOR ORCIDs

S. Isabel  http://orcid.org/0000-0001-7277-4150
A. Eshaghi  http://orcid.org/0000-0001-5150-483X
J. B. Gubbay  http://orcid.org/0000-0003-0026-3786
T. W. A. Braukmann  http://orcid.org/0000-0002-2452-3776

## AUTHOR CONTRIBUTIONS

S. Isabel, Conceptualization, Data curation, Formal analysis, Visualization, Writing – original draft, Writing – review and editing | A. Eshaghi, Conceptualization, Data curation, Formal analysis, Visualization, Writing – original draft, Writing – review and editing | V. R. Duvvuri, Conceptualization, Formal analysis, Writing – original draft, Writing – review and editing | J. B. Gubbay, Writing – review and editing | K. Cronin, Formal analysis, Writing – review and editing | Aimin Li, Writing – review and editing | M. Hasso, Writing – review and editing | S. T. Clark, Writing – review and editing | J. P. Hopkins, Writing – review and editing | S. N. Patel, Writing – review and editing | T. W. A. Braukmann, Conceptualization, Data curation, Formal analysis, Visualization, Writing – original draft, Writing – review and editing

## DATA AVAILABILITY

Genome sequences were deposited in NCBI with accession numbers OR759145 to OR759235.

## ETHICS APPROVAL

This project did not require research ethics committee approval as the activities described in this manuscript were conducted in fulfillment of Public Health Ontario's legislated mandate "to provide scientific and technical advice and support to the health care system and the Government of Ontario in order to protect and promote the health of Ontarians" (Ontario Agency for Health Protection and Promotion Act, SO 2007, c 10) and are therefore considered public health practice, not research. The sequences generated using this approach were used to support outbreak and routine surveillance of mpox in Ontario.

## ADDITIONAL FILES

The following material is available online.

## Supplemental Material

**Figure S1 (Spectrum02979-23-s0001.pdf).** Maximum-likelihood tree of WGS of monkeypox virus specimens from Ontario and references with bootstrap values greater than 70 indicated at nodes.
**Supplemental tables (Spectrum02979-23-s0002.xlsx).** Tables S1, S2, and S3.

## Open Peer Review

**PEER REVIEW HISTORY**

**(review-history.pdf).** An accounting of the reviewer comments and feedback.

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
