## [Reviewer comments · Microbiology Spectrum]

Microbiology Spectrum

Targeted Amplification Based Whole Genome Sequencing of Monkeypox virus in Clinical Specimens

S. Isabel, A. Eshaghi, V. R. Duvvuri, J. B. Gubbay, K. Cronin, Aimin Li, M. Hasso, S. T. Clark, J. P. Hopkins, S. N. Patel, T. W. A. Braukmann

Corresponding Author(s): T. Braukmann, Public Health Ontario

Review Timeline:

Submission Date:	September 5, 2023
Editorial Decision:	October 4, 2023
Revision Received:	October 20, 2023
Accepted:	October 29, 2023

Editor: Heba Mostafa

Reviewer(s): The reviewers have opted to remain anonymous.

Transaction Report:

DOI: <https://doi.org/10.1128/spectrum.02979-23>

October 4, 2023

Dr. Thomas Werner Anthony Braukmann
Public Health Ontario
Microbiology and Laboratory Science
661 University Ave
Toronto, ON M5G 1M1
Canada

Re: Spectrum02979-23 (Targeted Amplification Based Whole Genome Sequencing of Monkeypox virus in Clinical Specimens)

Dear Dr. Thomas Werner Anthony Braukmann:

Thank you for submitting your manuscript to Microbiology Spectrum. As you will see your paper is very close to acceptance. Please modify the manuscript along the lines I have recommended. As these revisions are quite minor, I expect that you should be able to turn in the revised paper in less than 30 days, if not sooner. If your manuscript was reviewed, you will find the reviewers' comments below.

When submitting the revised version of your paper, please provide (1) point-by-point responses to the issues raised by the reviewers as file type "Response to Reviewers," not in your cover letter, and (2) a PDF file that indicates the changes from the original submission (by highlighting or underlining the changes) as file type "Marked Up Manuscript - For Review Only". Please use this link to submit your revised manuscript. Detailed instructions on submitting your revised paper are below.

Link Not Available

Sincerely,

Heba Mostafa

Reviewer comments:

Reviewer #1 (Comments for the Author):

response to reviewers adequate addresses reviewer concerns.

Reviewer #2 (Comments for the Author):

Isael and colleagues describe a targeted amplification approach for the detection and sequencing of monkeypox genomic material from clinical specimens. This reviewer thanks the authors for their thoughtful responses and inclusions to the previously submitted queries - the additional technical information provides context for the issues raised, particularly the discussion concerning the use of longer versus shorter amplicon sizes as reported in Chen et al.

Lines 205-215. In order to present a more balanced assessment, what are the drawbacks to a tiled targeted-amplification approach compared to WGS/metagenomics? ...perhaps with respect to mutation leading to loss of amplification as alluded to in Lines 226-228? When would it not be appropriate to consider this type of approach? There is good discussion concerning the advantages of this method, but it is also important to briefly explore limitations and caveats.

Would suggest breaking the Discussion section into paragraphs like the Introduction. There are multiple locations in text where this can be easily accomplished (i.e. Line 215, 220, etc.) These do appear in the marked-up version of the manuscript, but not in the text file.

It is appreciated that the authors added text concerning hybrid bait capture at the request of the reviewers, although its placement at the end of the manuscript feels very disjointed and out of place. Would suggest finding a more appropriate place in the manuscript for a smoother incorporation of this comparison.

Preparing Revision Guidelines

Please return the manuscript within 60 days; if you cannot complete the modification within this time period, please contact me. If you do not wish to modify the manuscript and prefer to submit it to another journal, please notify me of your decision immediately so that the manuscript may be formally withdrawn from consideration by Microbiology Spectrum.

Reviewer comments:

Reviewer #1 (Comments for the Author):

response to reviewers adequate addresses reviewer concerns.

Thank you for your review of our manuscript, we appreciate your time and effort in improving our manuscript.

Reviewer #2 (Comments for the Author):

Isael and colleagues describe a targeted amplification approach for the detection and sequencing of monkeypox genomic material from clinical specimens. This reviewer thanks the authors for their thoughtful responses and inclusions to the previously submitted queries - the additional technical information provides context for the issues raised, particularly the discussion concerning the use of longer versus shorter amplicon sizes as reported in Chen et al.

Thank you for your review of our manuscript, we appreciate your time and effort in improving our manuscript. We agree with your additional comments. We have added some caveats or limitations of using primer based approaches for sequencing viral genomes in the discussion section. We have also rearranged our discussion to improve flow and structure of the text.

Lines 205-215. In order to present a more balanced assessment, what are the drawbacks to a tiled targeted-amplification approach compared to WGS/metagenomics? ...perhaps with respect to mutation leading to loss of amplification as alluded to in Lines 226-228? When would it not be appropriate to consider this type of approach? There is good discussion concerning the advantages of this method, but it is also important to briefly explore limitations and caveats.

We have added additional text discussing the limitations and caveats of primer based approaches in the discussion section.

Would suggest breaking the Discussion section into paragraphs like the Introduction. There are multiple locations in text where this can be easily accomplished (i.e. Line 215, 220, etc.) These do appear in the marked-up version of the manuscript, but not in the text file.

We have broken up the discussion into discrete paragraphs to help with the reading flow.

It is appreciated that the authors added text concerning hybrid bait capture at the request of the reviewers, although its placement at the end of the manuscript feels very disjointed and out of place. Would suggest finding a more appropriate place in the manuscript for a smoother incorporation of this comparison.

The text on hybrid bait capture was moved earlier in the discussion section. We have improved the organization and flow of manuscript.

Re: Spectrum02979-23R1 (Targeted Amplification Based Whole Genome Sequencing of Monkeypox virus in Clinical Specimens)

Dear Dr. Thomas Werner Anthony Braukmann:

Your manuscript has been accepted, and I am forwarding it to the ASM production staff for publication. Your paper will first be checked to make sure all elements meet the technical requirements. ASM staff will contact you if anything needs to be revised before copyediting and production can begin. Otherwise, you will be notified when your proofs are ready to be viewed.

Sincerely,
Heba Mostafa
Editor
Microbiology Spectrum